# Comparative Evaluation of Different Surface Coatings of Fe_3_O_4_-Based Magnetic Nano Sorbent for Applications in the Nucleic Acids Extraction

**DOI:** 10.3390/ijms23168860

**Published:** 2022-08-09

**Authors:** Anna Szymczyk, Marcin Drozd, Agnieszka Kamińska, Magdalena Matczuk, Maciej Trzaskowski, Marta Mazurkiewicz-Pawlicka, Robert Ziółkowski, Elżbieta Malinowska

**Affiliations:** 1Chair of Medical Biotechnology, Faculty of Chemistry, Warsaw University of Technology, Stanisława Noakowskiego 3, 00-664 Warsaw, Poland; 2Doctoral School No. 1, Warsaw University of Technology, Plac Politechniki 1, 00-661 Warsaw, Poland; 3Centre for Advanced Materials and Technologies CEZAMAT, Warsaw University of Technology, Poleczki 19, 02-822 Warsaw, Poland; 4Chair of Analytical Chemistry, Faculty of Chemistry, Warsaw University of Technology, Stanisława Noakowskiego 3, 00-664 Warsaw, Poland; 5Faculty of Chemical and Process Engineering, Warsaw University of Technology, Ludwika Waryńskiego 1, 00-645 Warsaw, Poland

**Keywords:** iron(II,III) oxide, magnetic nanoparticles synthesis, Fe_3_O_4_ MNPs functionalization, DNA extraction, DNA—nanoparticle interactions

## Abstract

Nucleic acid extraction and purification are crucial steps in sample preparation for multiple diagnostic procedures. Routine methodologies of DNA isolation require benchtop equipment (e.g., centrifuges) and labor-intensive steps. Magnetic nanoparticles (MNPs) as solid-phase sorbents could simplify this procedure. A wide range of surface coatings employs various molecular interactions between dsDNA and magnetic nano-sorbents. However, a reliable, comparative evaluation of their performance is complex. In this work, selected Fe_3_O_4_ modifications, i.e., polyethyleneimine, gold, silica, and graphene derivatives, were comprehensively evaluated for applications in dsDNA extraction. A family of single batch nanoparticles was compared in terms of morphology (STEM), composition (ICP-MS/MS and elemental analysis), surface coating (UV-Vis, TGA, FTIR), and MNP charge (ζ-potential). ICP-MS/MS was also used to unify MNPs concentration allowing a reliable assessment of individual coatings on DNA extraction. Moreover, studies on adsorption medium (monovalent vs. divalent ions) and extraction buffer composition were carried out. As a result, essential relationships between nanoparticle coatings and DNA adsorption efficiencies have been noticed. Fe_3_O_4_@PEI MNPs turned out to be the most efficient nano sorbents. The optimized composition of the extraction buffer (medium containing 0.1 mM EDTA) helped avoid problems with Fe^3+^ stripping, which improved the validity of the spectroscopic determination of DNA recovery.

## 1. Introduction

Seeking the most accessible molecular diagnostic tests and developing reliable and cheap procedures is of utmost importance [1]. In most cases, nucleic acid analysis’s first step is their extraction and purification from biological material. To assure the reliability of detection assays, such as qRT-PCR or genosensors, it is crucial to obtain high-quality purified DNA or RNA [2]. Classical nucleic acid isolation procedures include phenol-chloroform extraction, salting out and proteinase K treatment, and adsorption on silica–gel membrane [3]. These efficient but time-consuming methods require hazardous solutions, dedicated equipment, and expensive consumables [4]. The newly developed procedures should be fast, have throughput, and involve the minimum possible number of reagents or benchtop equipment. Considering the abovementioned issues, nanomaterials gained particular attention as an indispensable part. Their properties, like high surface-to-volume ratio and the easiness of surface properties adjustment, became the reason for their applications, e.g., in genetic material extraction and purification [5,6,7,8]. Carefully chosen magnetic nanoparticle (MNPs) surfaces can show a high affinity for nucleic acids. In contrast, the presence of a magnetic core allows them to be easily manipulated using an external magnetic field. The DNA should bind preferentially to nanoparticles which facilitate its removal even from complex samples (i.e., proteins or other compounds). Then the application of external magnetic force separates the MNPs. Further changing the solution where MNPs are dispersed induces medium-triggered DNA adsorption reversibility, ensuring its facile recovery back to the solution.

Current research trends targeted at new nano sorbents for the extraction of nucleic acids focus on the development of both magnetic core materials, their shape and morphology [9], and their surface functionalization [10,11]. Several magnetic materials based on iron, cobalt, and nickel have been developed, e.g., magnetic cobalt–zinc ferrite core/SiO_2_ shell nano sorbents [12], cobalt-oxide-based nanoparticles [13], spinel iron-cobalt oxide compounds [14], hydrophobic magnetic deep eutectic solvents containing Fe/MnCo/Gd ions [15], and magnetic ionic liquids including cobalt(II) and nickel(II) complexes [16]. Cobalt ferrite and other types of novel magnetic nanoparticles are promising due to their magnetocrystalline anisotropy, good parameters of coercivity, and saturation magnetization. Such materials also offer high chemical stability, wear resistance, and generally high physical and chemical stability [17]. The latest concepts in designing magnetic nano sorbents also include the use of magnetic nanoparticle (MNP) assemblies [9], the use of porous structures [18], or the use of specific double helix formation interactions [19]. However, due to the low cost and simplicity of synthesis, magnetite remains the dominant magnetic material in magnetic nanoparticles dedicated to nucleic acid extraction and purification.

The current scientific literature does not indicate a universal type of surface modification and the conditions that ensure efficient nucleic acid extraction. Therefore, there are several concepts involving different kinds of interactions, including electrostatic attraction between the nanoparticle and polyanionic phosphate backbones of nucleic acids [20,21], hydrophobic interactions (*π-**π* stacking) [22], hydrogen bonds formation [23], coordination and salt bridging [24] or specific, biological affinity [25,26]. Nevertheless, as could be found in the literature also, bare Fe_3_O_4_ nanoparticles, without clearly described surface properties, were successfully employed for DNA isolation [27]. Recently, Qi et al. proposed a method of capturing DNA adducts from human blood samples through Fe_3_O_4_@GO nano sorbent, where GO was responsible for improving selectivity by enhancing the interaction with the analyte [28]. On the other hand, in the recent publication by Zhang, the high and pH-sensitive DNA loading capacity was due to cationic polyethyleneimine [29]. Paltrinieri et al. coated Fe_3_O_4_ MNPs with polyallylamine hydrochloride (PAH), and PAH functionalized with guanidinium groups (PAH–Gu) for enhanced phosphate binding [30]. An interesting and recently reported example of a polycationic ligand is PEDOT. Nanoparticles modified with this polymer showed a unique, high binding capacity in acidic media [21]. Moreover, silica coatings are commonly used for the solid-phase purification of DNA [31]. Silanization with modified precursors (APTES/MPTMS) allows easily manipulating the character of functional groups, which also affects affinity towards nucleic acids [32]. Min et al. described an approach to isolate and purify DNA based on hydrogen bonding via carboxyl groups. These superparamagnetic Fe_3_O_4_ nanoparticles were modified with *meso*-2,3-dimercaptosuccinic acid (DMSA) [33].

Despite the growing number of reported examples of magnetic nanomaterials as nucleic acid nano sorbents, relatively little attention is paid to the critical evaluation of adsorption mechanisms and their impact on the nucleic acids’ extraction capacity. The comparative studies so far focus on quantitative analysis of solid- and liquid-phase extraction [34,35] and the confrontation of manual and automatic approaches [36]. The variety of conditions and magnetic properties of the Fe_3_O_4_ cores make a comprehensive comparison of the effects of surface coating difficult. Therefore, the available literature examples concern mainly the comparison of bare and core-shell nanoparticles [37] or methods of selective and non-selective adsorption of nucleic acids [38].

Presented studies focus on the comparative analysis of the influence of the nanoparticle surface type on the efficiency of its interaction with calf thymus DNA. The proposed nanoparticle preparation method allowed several post-synthetic modifications of the starting Fe_3_O_4_@PEI nanoparticles. Fe_3_O_4_ surface modifications cover the functionalization of a magnetic core with polyethyleneimine (PEI) and further with graphene oxide (GO), carboxylated graphene oxide (GOCOOH), gold (Au), silica/amine-silica (TMOS/APTMS). For comparison, commercially available Fe_3_O_4_ nanoparticles without polymer coating were analyzed in the presented studies. Obtained Fe_3_O_4_ types (coatings in the form of solid shells as well as surface ligands) were extensively characterized in terms of their morphology, qualitative and quantitative elemental composition, and surface properties by STEM imaging, ICP-MS, and elementary analysis, thermogravimetry, ζ-potential measurements, UV-Vis- and Fourier-transform infrared spectra registering. Additionally, we comparatively characterized nanomaterials’ magnetic properties (specific magnetization and rate of magnetic separation) and determined the iron content in their prepared suspensions using ICP-MS/MS. The latter results allowed for the normalization of MNPs concentration (the same batch of magnetite core). They provided a reliable comparison of the influence of the nanoparticle surface type on the efficiency in its interaction with calf thymus DNA. The adsorption and desorption efficiency from the surface of modified nanoparticles was investigated by UV-Vis spectrophotometry. The effect of various Na^+^ or Mg^2+^ concentrations in the adsorption medium and the influence of temperature and concentration of the buffer components used in the DNA desorption protocols were also examined.

## 2. Results and Discussion

Magnetic nanoparticles’ properties, like good dispersion, high surface area, and ease of surface modification, make them ideal as sorbents, carriers for various biological species, also nucleic acids. Thus, they have broad applicability in developing the DNA/RNA analysis methods or as components in genetic material preparation assays (extraction and purification steps) [39]. The magnetic core has two essential functions. First, to provide the ability of rapid separation in an external magnetic field from the sample. Second, Fe_3_O_4_ acts as a platform for further functionalization with compounds expressing the affinity to nucleic acids. 

However, due to nanoparticles’ magnetic properties and the tendency to clusterization, maintaining their biological species adsorption efficiency typically requires their stabilization [40]. A stabilizer is also necessary to secure the Fe_3_O_4_ nanoparticles as a convenient platform for further modification with various coatings. The presented studies achieved the first use of polyethyleneimine (PEI) as a particle stabilizer introduced in situ during the MNPs co-precipitation synthesis method previously described by Zhou [41]. Typically, stabilization of PEI is accomplished by post-synthetic ligand exchange from the starting citrate-capped Fe_3_O_4_ nanoparticles. The already described post-synthetic modification takes more time (an additional 4 h), elevated temperature (80 °C), and strict control of pH during the process [42,43]. In situ capping of Fe_3_O_4_ nanoparticles with PEI significantly simplifies the procedure (one-pot approach) and helps maintain good colloidal stability of nanoparticles by eliminating the risk of aggregation during the ligand exchange of citrate-capped nanoparticles.

The presented research aimed to prepare magnetic nanoparticles of different surface modifications and their comparative analysis in nucleic acids (calf thymus dsDNA) adsorption/desorption process efficiency. For this purpose, seven types of surface coatings of magnetic MNPs that may interact with DNA in various ways were prepared and characterized. As was stated above, PEI provided good stabilization and gave the cationic character of the surface over a wide pH range. Based on magnetic nanoparticles coated with PEI (Fe_3_O_4_@PEI), a family of functionalized nanoparticles characterized by uniform cores was synthesized. This included: two types of silica shells, anionic silica (Fe_3_O_4_@TMOS) and cationic amine-silica (Fe_3_O_4_@APTMS); two types of graphene derivatives, graphene oxide (Fe_3_O_4_@GO) and carboxylated graphene oxide (Fe_3_O_4_@GOCOOH). To enable further covalent functionalization of magnetic nanoparticles with thiolated receptors, nanoparticles coated with gold shells (Fe_3_O_4_@Au) were synthesized. Additionally, for comparison with as-obtained nano sorbents, the commercially available MNPs without polymer shells were considered (the manufacturer did not provide information on the surface coating composition). We expected that in the case of commercial Fe_3_O_4_ nanoparticles, their core ligands (e.g., adsorbed anions) are directly responsible for the interaction with the nucleic acid. Schemes of nanoparticles used in the framework of this study are depicted in Figure 1.

It should be emphasized that before MNPs usage for nucleic acid interaction analysis, the concentration of nano sorbents on various surfaces was unified. It was achieved by using the same batch of MNPs for all modifications and the expression of nanoparticle concentrations by iron content (magnetite core). This step was crucial for the reliability of the efficiency of the comparative evaluation of DNA/MNPs interactions.

### 2.1. Evaluation of Magnetic Properties of Fe_3_O_4_-Based Nanosorbents

To assess the influence of surface modification on the rate of magnetic separation, we measured the time from external magnetic field application to obtain the complete MNPs collection. A series of photos before, during, and after applying an external magnetic field were taken, and exemplary images are shown in Figure 2a. In addition, magnetization curves were recorded for each nanoparticle type (selected examples shown in Figure 2b), and the saturation magnetization values were determined.

As can be seen, functionalized nanoparticles retain their ability to magnetic separation, but the rate of magnetic collection strongly depends on the type of surface coating. The rates of MNPs separation were compared to MNPs Fe_3_O_4_@PEI, which were used as an internal benchmark (up to 50 s). The fastest separation (up to 30 s) was observed for nanoparticles coated with silica, amine-silica, and commercial MNPs. This can be attributed to their sizes and morphology. In the case of Fe_3_O_4_@TMOS and Fe_3_O_4_@APTMS, partial agglomeration at the silanization stage typically occurs, resulting in large, multi-core structures. On the other hand, commercially available nanoparticles have a larger core diameter than nanoparticles obtained by co-precipitation, which is also reflected in their behavior in a magnetic field. Furthermore, nanoparticles decorated with GO and GOCOOH were characterized by good colloidal stability and rapid separation (up to 40 s) in the magnetic field compared to PEI-coated magnetic cores. Notably, when attached to GO and carboxylated graphene oxide, magnetic nanoparticles gain properties different from those obtained from other modifications. It was noticed that such nanoparticles tend to form macroscopic graphene-like flakes (row 3, column 2 in Figure 2a). Such behavior of the obtained magnetic nanostructures confirms the effectiveness of the Fe_3_O_4_ decoration with graphene oxide sheets. However, the tendency of such nanoparticles to agglomerate via π-π stacking does not deteriorate their applicability as nano sorbents, as they can be easily redispersed through sonication.

Macroscopic observations were confirmed by detailed magnetometric analysis. As shown in Figure 2b, coercivity values were low, indicating that synthesized MNPs show typical superparamagnetic properties. This is also evidenced by the relatively high saturation magnetization values (Figure 2c), which are slightly lower but comparable to the magnetization of pure magnetite (92 emu/g). As can be seen in the detailed diagrams presented in Figure 2b, the M(H) magnetization curves in a wide range of magnetic fields correlate with the saturation magnetization values. The similar nanoparticle types compared with each other show a similar course of the magnetization process (see Fe_3_O_4_@GO vs. Fe_3_O_4_@GOCOOH and Fe_3_O_4_@TMOS vs. Fe_3_O_4_@APTMS). The poor magnetizing ability of Fe_3_O_4_@Au compared to Fe_3_O_4_@PEI and Fe_3_O_4_ (commercial) is also confirmed. Obtained values in most cases were still high enough to employ MNPs as powerful nano sorbents. On the other hand, Fe_3_O_4_@Au core-shell nanoparticles show a poor capability for magnetic separation (up to 120 s) and relatively low magnetization saturation (44.6 emu/g). This phenomenon can be explained by the presence of a conductive Au shell, which significantly reduces the nanoparticle magnetism [41] due to the mass effect of the non-magnetic coating. This type of MNPs separation was ineffective; Au-coated MNPs were excluded from detailed studies on DNA adsorption/desorption.

### 2.2. ζ-Potential Measurements

ζ-potential parameter of synthesized MNPs has been used to portray the changes in their surface modification (Figure 3). From the family of studied nanoparticles, only Fe_3_O_4_@PEI (25.3 mV) and Fe_3_O_4_@APTMS (30.2 mV) are characterized by a positive charge due to the presence of protonated amino groups of the ligands. Commercial MNPs possess an intrinsic negative charge (−22.6 mV). The anionic character of these nanoparticles without polymer shells in near-neutral pH (different from the expected, intrinsically slightly cationic, bare iron(II,III) oxide) comes most likely from the complexation of the surface atoms by anions—either naturally occurring in the sample, or compounds added as a surface stabilizer (e.g., commonly used citrate) [44]. The negative charge of GO-decorated Fe_3_O_4_ (−16.3 mV) and GOCOOH-decorated Fe_3_O_4_ (−23.3 mV) confirms that originally cationic, PEI-coated iron(II,III) oxide nanoparticles are efficiently incorporated into the structure of graphene oxide. The intrinsic negative charge of GO and GOCOOH derives from carboxyl and other oxygen-containing groups on its surface. Fe_3_O_4_@TMOS (−32.0 mV) are negatively charged due to silanol groups of the silica shell. On the other hand, the negative charge of Fe_3_O_4_@Au (−16.5 mV) comes from a citrate ion, which acts as a stabilizer of gold nanoparticles. It can be concluded that all nanoparticles gained the expected charge due to their coatings, proving their ligand attachment effectiveness. 

### 2.3. MNPs Morphology (Bright-Field Scanning Transmission Electron Microscopy)

Another critical parameter influencing the MNP’s DNA sorption efficiency (e.g., described above silica or amine-silica coated) is their morphology. For this reason, the family of nanoparticles has been characterized by STEM. Average nanoparticle sizes were investigated based on the analysis of the obtained micrographs (Figure 4). Despite the same sizes of magnetic cores (except MNPs shown in Figure 4g), nanoparticles differ significantly in terms of their morphology, which is caused by surface modification. Fe_3_O_4_@PEI are characterized by the smallest dimensions (average diameter 8.1 nm—Figure 4a) compared to other structures. Decoration of magnetic MNPs with graphene oxide (b,c), solid silica, or amine-silica shells (d,e) also can be observed in micrographs. Fe_3_O_4_@PEI nanoparticles have been successfully adsorbed on the surface of GO and GOCOOH by electrostatic self-assembly. Moreover, compared to Fe_3_O_4_@GO (magnetic core size 8.8 nm), Fe_3_O_4_@PEI are wrapped more closely at the GOCOOH flake surface (core size 9.0 nm). Based on the core sizes, the decoration process did not affect the average diameter concerning Fe_3_O_4_@PEI. The formation of silica coatings significantly influences nanoparticle morphology, Fe_3_O_4_@TMOS, and Fe_3_O_4_@APTMS. In both cases, larger structures are formed (34.4 and 165.1 nm, respectively), which may be explained by the occlusion of several Fe_3_O_4_ cores inside a single silica shell. On the other hand, coating MNPs with solid gold results in a slight increase in diameter to 15.7 nm, and no significant agglomeration of nanoparticle cores were visible (Figure 4f). The only nanoparticles with originally different core diameters are commercial Fe_3_O_4_ nanoparticles (16.4 nm) shown in Figure 4g.

### 2.4. Compositional Analysis by Thermogravimetric Analysis (TGA)

Further characterization of the obtained core-shell nanoparticles in terms of stability and composition involved thermogravimetric analysis. The obtained thermograms show that the MNPs samples differ in thermal stability (Figure 5). TGA analysis shows that the most stable samples are those covered with a solid layer of gold. The course of thermogram of Fe_3_O_4_@Au MNPs is similar to parallelly co-precipitated NPs using the same protocol but without the addition of an external stabilizer (such NPs were not examined as nano sorbents in this study and used only for comparative characterization). The nanoparticles covered with a solid gold layer are characterized by the slightest loss of mass, indicating a small contribution of organic matter in their structure. In the case of silane-coated NPs, the oxidation temperature was lower than for the other samples. It can be concluded from the course of the TGA curve that both types of silane coatings have entrapped a higher amount of water than the other shell materials. APTMS-based coating exhibit the most significant weight loss, demonstrating its lower stability compared to TMOS-based silica. Slow oxidation of organic occlusions entrapped in a lattice of aminated silica is visible as the mass loss curve, which does not flatten despite the increase in temperature up to 750 °C. The significant mass loss is visible for MNPs modified with both graphene derivatives, which proves a substantial contribution of organic matter in the final product. Among the MNPs decorated with GO and GOCOOH, a minor loss is observed for the GOCOOH sample. This may be due to the lower water content and a slight reduction of GOCOOH compared to GO. A slight but noticeable weight loss (compared to unmodified MNPs) was observed for PEI-coated MNPs, which proves the effective stabilization of Fe_3_O_4_@PEI. 

### 2.5. FT-IR Spectroscopy 

The successful coating of Fe_3_O_4_ MNPs with various shells was also confirmed by FT-IR spectroscopy. In Figure 6, the FT-IR spectra of modified nanoparticles are presented and compared with the spectra of corresponding modifiers in a pure form. As can be easily noticed, the spectra of surface-modified magnetic nanoparticles manifest the characteristic bands derived from the related modifiers.

In all nanoparticle cases, the strong band derived from Fe-O is noticeable, around 500 cm^−1^, characteristic of metal-oxygen bonds. Although Fe_3_O_4_@PEI nanoparticles have been coated with a polymer, as evidenced by previous TGA and ζ-potential analysis, no characteristic bands assigned to the PEI polymer bonds can be observed in Figure 6a. This can be explained by a relatively thin molecular layer of ligands and thus their low amount in the analyzed sample. Hence, Au and PEI-coated MNPs do not differ significantly in the course of the spectrum from uncoated MNPs. On the other hand, clear bands corresponded to GO or TMOS modifiers shown in Figure 6b,c may indicate a higher amount deposited at the MNPs surface. Spectra of surface-modified magnetic nanoparticles show specific bands in the exact wavenumber ranges as pure materials. For Fe_3_O_4_@silica, characteristic bands of Si-O and Si-O-Si bonds are noticeable, which corresponds well to TMOS precursor (except C-H bonds from methyl residuals) C-H. For Fe_3_O_4_@GO and GO, similar bands assigned to C=O, C-O, and primary and secondary O-H bands, have also been identified. The characteristic bands evidenced the presence of the carboxylate group: at ~1600 cm^−1^, the band attributed to the vibration COO^−^ (v_as_), and in the range 1300 cm^−1^–1400 cm^−1^, the bands attributed to the vibrations COO^−^ (v_s_)– and CO (v_s_) + OCO (δ). The bands (COO^−^ (v_s_) are more leading for pure GO in removing Fe_3_O_4_@GO.

### 2.6. Analysis of the Nanoparticles Composition

The elemental and ICP-MS/MS analyses were employed to determine the content of the nanoparticle’s main elementary components. As seen in Table 1, and as expected, the dominant component element of MNPs is iron oxide, represented by the determined iron and estimated oxygen contents (based on the Fe/O molar fraction for nanoparticles without stabilizer). At the same time, the presence of different coatings was reflected by the contributions of different hetero-elements. In the case of nanoparticles decorated with organic coatings (PEI, GO, and GOCOOH), carbon was the dominant hetero-element. Due to the presence of polyethyleneimine in all types of particles (except bare MNPs), a certain proportion of nitrogen was observed, coming from the amine groups of the polymer. At the same time, the GO and GOCOOH decorated nanoparticles have a significantly higher ratio of carbon to nitrogen (4.11 for GO and 3.94 for GOCOOH, respectively) compared to PEI coated NPs (1.35). The shells of Fe_3_O_4_@TMOS and Fe_3_O_4_@APTMS MNPs turned out to be much thicker than the remaining ones—the contribution of silicon atoms was in both cases above 40% (*w*/*w*) and was higher than the iron content. The unmodified silica obtained with TMOS coating was characterized by a low carbon and nitrogen content in contrast to the APTMS coating, in which numerous ω-aminopropyl residues have been trapped. The significant gold coverage of Fe_3_O_4_@Au magnetic nanoparticles was also confirmed. The proportion of this element at 35.83% by mass is slightly lower than that of Fe as the main core constituent (43.21%). The obtained results unequivocally confirm the modification procedures’ effectiveness for all subjected to test nanoparticles.

### 2.7. Standardization of Magnetic Nano Sorbents Concentration

The total iron content in MNPs samples was determined using the ICP-MS/MS technique to standardize the dose of DNA nano sorbents taken into further tests. The obtained iron concentrations were used to calculate the amount of MNPs unequivocally. The idea was based on the assumption that all nanoparticle modifications were performed by directly comparing the nucleic acid binding efficiency between variously modified MNPs. As a result, the concentration of nanoparticles in all samples was unified to a total Fe concentration of 72.96 ± 0.08 µg/mL (corresponding to the lowest concentration of iron in the undiluted MNPs sample). Based on the above data, appropriate sample solutions were prepared for further DNA adsorption and recovery studies to ensure the same concentration of magnetic cores within all experiments.

### 2.8. UV-Vis Absorption Spectra Analysis

The formation of surface coating usually entails a change in the optical properties of MNPs, which results in increased ligand absorption and scattering of the core-shell type structure. Absorption spectra shown in Figure 7 confirm the attachment of corresponding ligands or the formation of solid shells on the surface of magnetic cores. A slight band around 230–240 nm observed for Fe_3_O_4_@GO and Fe_3_O_4_@GOCOOH MNPs can be attributed to π-π* transitions in graphene structure [45]. On the other hand, Fe_3_O_4_@TMOS and Fe_3_O_4_@APTMS MNPs show a high scattering, typical for larger structures [46]. In turn, the significant absorption at approximately 520–550 nm by Fe_3_O_4_@Au MNPs can be explained by the occurrence of surface plasmon resonance [47] (solid dark blue line). The results of the UV-Vis spectra analysis are generally consistent with the conclusions of the STEM and ζ-potential studies.

### 2.9. Studies of DNA Interactions with Modified MNPs

For the quantitative evaluation of calf thymus DNA adsorption and desorption processes, a spectrophotometric method based on the specific absorption wavelength of nucleobases at 260 nm was used [29]. The efficiency of the above steps was calculated as Δ*A*. It refers to the ratio [%] of DNA amounts in media before adsorption and after nano sorbent separation (Figure 8).

#### 2.9.1. DNA Adsorption Studies

The efficiency of DNA adsorption can be influenced by several factors such as the nanoparticle surface and ligand type (solid shell, branched polymer), its charge (cationic or anionic), and the concentration of nanoparticles or medium ionic strength. As we standardized the concentration of nanoparticles in the sample, the primary influence on the obtained results should originate only from the type of MNPs modification. This allowed us to observe and compare the effect of surface coating on calf thymus DNA interactions efficiency with modified MNPs. As the nanoparticle residuals in the analyzed sample could influence the registered absorbance value (strong absorption in the UV range, Figure 7), it was indispensable to provide their quantitative separation. Therefore, only MNPs capable of rapid and complete separation were used for this study. Fe_3_O_4_@Au, which expressed slow separation kinetics and low efficiency, were excluded. Nonetheless, after the magnetic separation of all tested nanoparticles from the DNA solution, a significant loss of absorbance at 260 nm was observed. This change manifests a relatively high DNA binding capacity of prepared magnetic nano sorbents. In the case of cationic nano sorbent, Fe_3_O_4_@PEI, a high adsorption ratio (represented by Δ*A* values near 99%) was observed even in the medium of low ionic strength (Figure 9a). Together with increasing the Na^+^ or Mg^2+^ concentration, this efficiency diminishes to even ~37% when calf thymus DNA was dissolved in an aqueous solution containing 1M Mg^2+^. This can be explained that the increase of the medium ionic strength can result in stronger DNA charge shielding, which adversely affects the adsorption process due to the electrostatic attraction. The observed results are in accordance with the postulated interaction mechanism, in which the positively charged surface of nanoparticles attracts negatively charged DNA structure.

As shown in Figure 9c,d, the opposite trend to Fe_3_O_4_@PEI in solution ionic strength influence on the dsDNA adsorption can be observed for anionic nanoparticles decorated with graphene oxide and its carboxy-derivative. In this case, the main driving force for DNA adsorption is most likely π-π bonds formation with aromatic rings of carbon nanomaterial surface. However, the negative charge present at its surface can efficiently decrease nucleic acid attraction and adsorption. Nonetheless, a high concentration of cations in a binding medium can reduce the Debye Length of carbon nanomaterials. Its effectively attenuates electrostatic repulsion between DNA and nanoparticles. As observed, the adsorption efficiency increases when the medium ionic strength increase, from 32 to 96% for Fe_3_O_4_GO and 22 to 99% for Fe_3_O_4_@GOCOOH (Figure 9c,d). At a salt (Na^+^ and Mg^2+^) concentration of 0.5 M, both nanoparticle types showed almost quantitative adsorption of DNA, while the adsorption efficiency in water was significantly lower, approximating 30% for Fe_3_O_4_@GO and 20% for Fe_3_O_4_@GOCOOH. It may be explained by the slight difference in surface charge density (see ζ-potential studies) of both GO derivatives. As was mentioned above, the dominant mechanism for DNA adsorption on graphene oxide (and its derivative) is hydrophobic π-π stacking between aromatic rings. However, the hydrogen bond formation and donor-acceptor interactions (between oxygen-bearing moieties of GO and DNA nucleobases) should also be considered. In this point of view, GOCOOH provides oxygen functional groups that most likely can additionally interact with DNA (slightly lower adsorption efficiency in water).

In the case of silica-coated nanoparticles, both with positive (Fe_3_O_4_@APTMS) and negative (Fe_3_O_4_@TMOS) surface charge, a moderate ability to DNA binding (not exceeding 58% for Fe_3_O_4_@APTMS and 47% for Fe_3_O_4_@TMOS) was observed (Figure 9e,f). Moreover, for the above efficiency, the ionic strength of the solution was of rather limited influence. We expect that the higher the magnetic core agglomeration during their modification, the smaller the specific surface area available further for DNA adsorption of such nanoparticles, especially versus those where the cores agglomeration was not observed. Therefore we can note weaker adsorption efficiency for large, multi-core MNPs nanostructures coated with silica and amino-silica. The above observations can be explained because the formed, modified magnetic nanoparticles are multi-core constructs (Figure 4e,f). For this reason, the overall specific area available for DNA adsorption for Fe_3_O_4_@APTMS and Fe_3_O_4_@TMOS nanoparticles is highly reduced compared to other investigated single-core MNPs. The last investigated nano sorbent, commercial non-encapsulated nanoparticles, distinguishes the most significant sensitivity to the composition of the adsorption medium—a strong positive effect of divalent cations can be noticed (adsorption ratio increase from 54% for 0 M to 98% for 1 M MgCl_2_). However, what is of particular interest in the case of monovalent cations, adsorption decreases with increasing ionic strength (from 0 M to 1 M NaCl) (Figure 9b). As there is no precise information from the manufacturer regarding nanoparticle surface composition, the obtained results can be explained by the hydration or anion association on the Fe_3_O_4_ surface, which gained an anionic and polar character.

In general, the DNA extraction efficiency within this study may be influenced by factors such as (i) nanoparticle active surface area—long DNA chains may have a problem attaching to a nanoparticle with very small size; (ii) surface charge—DNA binding by cationic nanoparticle can be driven by electrostatic interactions; (iii) dispersion—the real surface area of MNPs per concentration unit may vary after modification with different coatings, e.g., multi-core particles or their aggregates; (iv) surface coatings types, like in the case of graphene oxide and possible *π-π* interactions with DNA.

Because of magnetic core concentration standardization in our study, the dominant factor influencing the extraction efficiency is the type of Fe_3_O_4_ surface coating, its charge, expressing possible interactions, and dispersion degree. The effect of surface charge is visible in the example of cationic nanoparticles coated with polyethyleneimine (characterized by very good adsorption due to electrostatic attraction of anionic DNA backbones). On the other hand, multi-core and aminated silica-coated MNPs are characterized by a poor DNA extraction efficiency due to the limited surface area available for adsorption. 

Na^+^ and Mg^2+^ ions influence both binding and desorption processes. Their presence is associated with ion bridging and charge screening during the interaction of DNA and nanoparticle surface. On the other hand, during the desorption process of nucleic acids from modified MNPs, their complexation by EDTA may play a significant role.

#### 2.9.2. Effect of EDTA Concentration on the Obtained Reliability of UV-Vis Results

Magnetic nanoparticles are investigated from the point of view of their application for nucleic acid adsorption, extraction, and its further desorption back to the solution. Both these steps, adsorption and desorption, take place in different environments. Typically, buffered and slightly alkaline media (pH~8.0) containing salt and chelating agents should be used for DNA desorption from nanoparticles [26,37]. The most common chelating agent is EDTA, while there are some discrepancies in the literature regarding its appropriate concentration, where typically 0.1–10 mM is used [26,37,48,49]. Moreover, the presence of EDTA in the desorption medium can have pros and cons. Apart from inhibiting nucleic acids restriction enzymes (DNases, RNases) and facilitating nucleic acid desorption (destabilization of coordination bonds and salt bridging), it can also negatively influence the amplification efficiency of nucleic acids. This was a motivation to initially use EDTA in a desorption medium at 1 mM. However, it had to be diminished even to 0.1 mM. Figure 10 shows the influence of EDTA concentration in desorption media on obtained results (for Fe_3_O_4_@PEI nanoparticles as an example). Moreover, the UV-Vis spectra of desorption buffer with two different EDTA concentrations and formed complex of EDTA with Fe^3+^ are presented.

First experiments using 10 mM TRIS-HCl of pH 8.0 and 1 mM EDTA as desorption media resulted in unexpectedly high DNA recovery (%), indicating the increase in nucleic acid concentration in the solution after its adsorption and desorption steps (Figure 10a). These repeated results raised doubts about the validity of the spectrophotometric method of DNA concentration determination in the desorption procedures [29,50]. Similar experiments for all nanoparticles without initially adsorbed calf thymus DNA showed a significant increase in UV-Vis absorbance for 260 nm. This confirmed that EDTA in the concentration of 1 mM has a crucial impact on the accuracy of desorption process evaluation using the established UV-Vis method. The reason was probably related to EDTA-induced stripping of Fe^3+^ from the magnetite core. The effect was additionally confirmed on UV-Vis spectra registered for 10 mM TRIS-HCl of pH 8.0 with EDTA (1 mM or 0.1 mM) solution with or without the addition of 100 µM FeCl_3_ (Figure 10b). As clearly seen, EDTA forms a complex with Fe^3+^ ion with a strong absorption band at around 260 nm (analytical wavelength in desorption analysis). Notably, a 10-fold decrease of the chelating agent concentration (to 0.1 mM) thoroughly eliminated its interference considering all types of MNPs (Figure 8a, Fe_3_O_4_@PEI as example). The background absorbance of supernatants was negligible, only slightly more significant than in the case of the complete absence of EDTA in the desorption medium. Therefore, for further studies on the DNA extraction process, TRIS buffer with 0.1 mM EDTA was used [50]. Such concentration seems to be a trade-off between ensuring the stability of the nucleic acid during the extraction process (by inhibiting nucleases) and suppressing interferences during spectrophotometric DNA determination.

#### 2.9.3. DNA Desorption and Recovery Studies

The final step of the presented investigations covers evaluating the calf thymus DNA recovery from the surface of the analyzed nanoparticles. UV-Vis spectra registered for solution (optimized as described above) after desorption and magnetic separation of nanoparticles enabled the assessment of DNA recovery concerning the amount of DNA adsorbed (100%) on nanoparticles (Figure 11). The desorption process was examined in various conditions, starting from mild (room temperature, short, 5-min incubation), gradually moving to more drastic (30-min incubation at 70 °C).

The 3D graphs presented in Figure 11 show the DNA recovery ratios at its individual desorption stages (*Z*-axis) versus different ionic strengths of the adsorption media (*X*-axis). The desorption studies were carried out for nanoparticles used in adsorption experiments in media of various ionic strengths. The cumulative desorption efficiencies for individual media with different salt contents are presented as corresponding insets in Figure 11.

The mono- and divalent cations (Na^+^ and Mg^2+^) influence several factors during DNA interaction with surfaces, e.g., charge screening, interactions with DNA backbone, dsDNA structure stabilization, or with chelating agents. These result from valence, polarizability, and EDTA chelating capacity differences. The DNA sugar-phosphate backbone shows a strongly anionic character in media with a pH close to neutral. Moreover, investigated surfaces express different charges, positive or negative. Multivalent electrolytes, in particular magnesium, are the most effective in charge screening and formation of salt bridges between adjacent anions [51,52]. Monovalent cations can bind to the DNA backbone as counterions neutralize the negatively charged DNA surface without bridging [53]. Cations’ co-adsorption facilitates DNA binding on the modified surfaces, which also possess a negative charge [54]. It was observed that in the case of the electrostatic mechanism of DNA interaction with cationic nanoparticles (Fe_3_O_4_@PEI and Fe_3_O_4_@APTMS), the beneficial effect of magnesium ions is more visible than for nanoparticles characterized by negative surface charge (lack of desorption in 1 M NaCl and 68% and 34% in 1 M MgCl_2_, respectively). In previously published works, it has been confirmed that the divalent ions support the formation of dense and more rigid DNA adsorbates due to the increased charge shielding and salt-bridging effect [51]. What is more, the facility of Mg^2+^ ion chelation by EDTA may favor the degradation of nanoparticle-DNA assemblies at the desorption step. Similarly, silica-coated nanoparticles exhibit slightly better DNA recoveries (from 0 to 42% for Fe_3_O_4_@APTMS and from 0 to 67% for Fe_3_O_4_@TMOS, respectively) when co-adsorbed with magnesium ions, which is consistent with a few reports in which both flat surfaces and SiO_2_-coated nano sorbents were examined [52,55].

Nanoparticles decorated with graphene derivatives (GO and GOCOOH), which interact with DNA primarily by attracting aromatic rings or donor-acceptor bonds, exhibit substantially different behavior in the DNA desorption process. In their case, magnesium ion did not promote DNA release. Additionally, significantly better DNA recovery ratios (up to 52% for Fe_3_O_4_@GOCOOH and from 0 to 78% for Fe_3_O_4_@GO) can be observed in more aggressive desorption conditions (long desorption times, elevated temperature), which indicates possibly slower desorption kinetics. A slight, beneficial effect of GO carboxylation on DNA desorption was also observed, presumably related to the increased content of polar, oxygen-containing groups available for hydrogen bonding. Commercial nanoparticles with larger diameters and without an additional shell should be considered separately. For such MNPs, the anionic and polar character of the surface is determined by the presence of hydroxyl groups and anions adsorbed on the bare Fe_3_O_4_ surface. The results obtained for these nanoparticles, e.g., relatively low DNA recoveries and good efficiency of its uptake, indicate a high nanomaterial affinity to DNA, resulting in its hindered desorption. Detailed results of cumulative desorption efficiencies (Insets in Figure 11) confirm the need for individual selection of optimal desorption parameters for each type of nanoparticle used as a DNA nano sorbent.

## 3. Materials and Methods

### 3.1. Reagents

Iron(III) chloride hexahydrate (FeCl_3_∙6H_2_O), iron(II) chloride tetrahydrate (FeCl_2_∙4H_2_O), 25% ammonium hydroxide (NH₄OH), branched polyethyleneimine (PEI) M_w_ = 25 kDa, hydrochloric acid (HCl), sodium hydroxide (NaOH), 96% ethanol (C_2_H_5_OH), tetramethyl orthosilicate (TMOS), sodium borohydride (NaBH_4_), gold(III) chloride trihydrate (HAuCl_4_∙3H_2_O), sodium citrate dihydrate (Na_3_C_6_H_5_O_7_∙2H_2_O), citric acid (HOC(COOH)(CH_2_COOH)_2_), hydroxylamine hydrochloride (NH_2_OH·HCl), hydrochloric acid (≥37%) and nitric acid (≥69.0%)—both for trace analysis, deoxyribonucleic acid from calf thymus, sodium chloride (NaCl), magnesium chloride hexahydrate (MgCl_2_∙6H_2_O), Trizma base (TRIS, NH_2_C(CH_2_OH)_3_ and ethylenediaminetetraacetic acid (EDTA, (HO_2_CCH_2_)_2_NCH_2_CH_2_N(CH_2_CO_2_H)_2_) were purchased from Sigma-Aldrich (St. Louis, MO, USA). 3-aminopropyl)trimethoxysilane (APTMS) was from Alfa Aesar (Haverhill, MA, USA). Graphene oxide (GO) was prepared by oxidation of natural graphite according to Hummer’s method [56] as described previously by Ziółkowski [45]. Carboxylated graphene oxide (GOCOOH) was prepared based on the methodology described by Song [57]. Iron(II,III) oxide nanoparticles suspension without polymer stabilizer, dispersed in water at 3 wt.%—further referred to as commercial Fe_3_O_4_ MNPs) was from PlasmaChem GmbH (Berlin, Germany). Iron, gold, and yttrium standard solutions were purchased from ULTRA Scientific, Inc (North Kingstown, RI, USA). High-purity water was produced by an Elix Water Purification system (Millipore, Molsheim, France) and used throughout this work.

### 3.2. Synthesis of Surface-Functionalized Magnetic Nanoparticles

In order to obtain a family of magnetic nanoparticles with various surface modifications, in the first stage Fe_3_O_4_@PEI nanoparticles were obtained by co-precipitation and then subjected to surface functionalization. Details of the experimental procedures are given in sections from Section 3.2.1. to Section 3.2.4, and the amounts of reagents used are summarized in Table 2.

#### 3.2.1. Fe_3_O_4_@PEI Magnetic Cores

Synthesis of iron(II,III) oxide nanoparticles was carried out using the modified co-precipitation method described by Zhou et al. using PEI as stabilizing ligand instead of sodium citrate [41]. The reaction was carried out at 90 °C under a nitrogen atmosphere with mechanical stirring at 2000 rpm. 4.886 g FeCl_3_∙6H_2_O and 2.982 g FeCl_2_∙4H_2_O were dissolved in 120 mL of water, and then 15 mL of a 25% ammonium hydroxide was quickly added. After 10 min, 20 mL of PEI aqueous solution (20 mg/mL) was injected. The stirring was carried out for the next 2 h. Then, the previously cooled suspension of Fe_3_O_4_ nanoparticles was washed five times by magnetic decantation, and the purified MNPs were reconstituted in 500 mL ultrapure water. For the separation of nanoparticles, cylindrical neodymium magnet (d = 70 mm, H = 50 mm) N42 was used. Magnetic nanoparticles coated with hyperbranched PEI were subjected to surface modification or used as obtained. A stock solution of Fe_3_O_4_@PEI nanoparticles with an optical density of OD_380_ = 1.5 in suitable solvents was used for further steps. MNPs without stabilizer (used for comparative purposes) were synthesized according to the same protocol but without the addition of PEI.

#### 3.2.2. GO and GO-COOH-Decorated Fe_3_O_4_

The procedures of decorating iron oxide(II,III) nanoparticles with GO and GOCOOH were carried out analogously. After 15 min sonication, 5 mL of Fe_3_O_4_ MNPs suspension (OD_380_ = 1.5) was added to 5 mL of GO/GOCOOH (10 mg/mL), and the whole mixture was magnetically stirred for 30 min. Then the nanoparticles were washed three times by magnetic decantation.

#### 3.2.3. Silica and Amine-Silica Coated Fe_3_O_4_

Silica-coated nanoparticles were prepared by the Störber method according to the modified procedure described by Z. Zhao [58]. Then, 150 mL of PEI-coated MNPs suspension (OD_380_ = 1.5) in anhydrous ethanol, 7.5 mL of deionized water, and 6 mL of ammonium hydroxide solution were mixed and sonicated for 30 min. 600 μL of TMOS was added into the mixture (in portions of 100 μL, with 10 min intervals of sonication) and vigorously stirred for the next 4 h at room temperature to allow the formation of silica layers. The obtained nanoparticles were washed three times by magnetic separation. To obtain nanoparticles coated with amine-silica, the mixture of TMOS and APTMS (3:1, *v*/*v*), instead of pure TMOS, was used, while the rest of the preparation procedure remained the same. 

#### 3.2.4. Fe_3_O_4_@Au Core-Shell Nanoparticles

Synthesis of spherical gold nanoparticles (Au seeds) was carried out with NaBH_4_ reduction of Au(III) using the method described by Brown et al. [59]. 10 mL 1% HAuCl_4_ solution and 900 mL of H_2_O were stirred magnetically. After 5 min, 20 mL of 38.8 M sodium citrate was added, followed by 4.5 mL of freshly prepared 0.075% NaBH_4_. The Au colloid solution was stirred for another 10 min. To prepare Fe_3_O_4_@PEI decorated with Au seeds, to 150 mL of Au nanoparticle’s suspension, 80 mL of a stock PEI-coated MNPs solution was added dropwise. The mixture was stirred for the next 4 h at room temperature. Washing was performed by magnetic decantation using 0.5 mM Na/citrate buffer pH 5.5. Then nanoparticles were suspended in 10 mM 250 mL NaOH for further Au deposition. To obtain Fe_3_O_4_@Au core-shell nanoparticles, a solid gold coating was deposited on the nanoparticles previously decorated with gold seeds by hydroxylamine-mediated HAuCl_4_ reduction. Au seeds-decorated Fe_3_O_4_@PEI, 100 μL of 10 mM HAuCl_4_, and 100 μL of 30 mM NH_2_OH∙HCl were added in 4 portions each at 3-min intervals with continuous stirring.

### 3.3. Nanoparticles Characterization

#### 3.3.1. Characterization of the Morphology, Magnetic and Surface Properties of MNPs

The magnetic properties of powdered MNPs samples were characterized by the SQUID vibrating sample magnetometer (QD-VSM) (Quantum Design). The suspension of nanoparticles was dried into powder form by magnetic collection, followed by drying the residual in a laboratory dryer at 60 °C until a stable mass of nanoparticle powder was obtained. The hysteresis loops were recorded at the varying magnetic field between −2.0 and +2.0 T at a constant temperature of 300 K stabilized to 0.02 K accuracy. For all types of nanomaterials, UV-Vis absorption spectra were recorded using a Lambda 25 spectrophotometer (Perkin Elmer, Waltham, MA, USA) and quartz microcuvette with a 1 cm pathlength (Hellma Analytics) in the range of 190–900 nm. ζ-potential of modified MNPs was measured by Zetasizer Nano ZS instrument (Malvern, UK) in disposable polystyrene cuvettes and dip cells with Pd electrodes. All ζ-potential measurements were carried out in three repetitions. BF- STEM nanoparticle micrographs were taken using Hitachi SU8230 ultra-high resolution field emission scanning-transmission electron microscope (Hitachi High-Technologies Corporation, Tokyo, Japan) at 30.0 kV accelerating voltage. Copper TEM grids coated with Lacey carbon film were, before observation, immersed in aqueous suspensions of MNPs and dried in ambient conditions.

#### 3.3.2. Determination of Total Iron Concentration in MNPs Samples

MNPs samples were homogenized (1 h of ultrasound bath) and digested using aqua regia in a Speedwave Four microwave-assisted digestion system (Berghof, Eningen, Germany) for 30 min at increased temperature (from 50 to 240 °C) and pressure (from 10 to 50 bar) with yttrium as internal standard (10 µg/L final concentration). The concentration of iron in samples was determined using an inductively coupled plasma tandem mass spectrometer working as an element-specific detector. The Agilent 8900 ICP Triple Quadrupole Mass Spectrometer (Tokyo, Japan) was equipped with a 2.5 mm quartz torch and the Pt-cones in the interface. The position of the torch and the nebulizer gas flow were adjusted daily, with emphasis paid to the increase the signal-to-noise ratio using a 1 µg/L solution of Co, Y, and Tl in 2% (*v*/*v*) HNO_3_ and 2% (*v*/*v*) HNO_3_, respectively. The RF power was 1550 W, nebulizer gas flow—1.09 L/min, reaction gas flow (hydrogen in ICP-MS/MS)—5.5 mL/min. The total concentrations of selected metal were calculated as a result of monitoring the mass/charge ratios 56 (^56^Fe), registered in the on-mass mode after the production in plasma and collision-reaction cell of singly-positively charged ions, and normalization (^89^Y) after daily external calibration against 10-point calibration lines (0–500 µg/L, R2 > 0.9998). 

#### 3.3.3. Qualitative and Quantitative Characterization of Various MNPs Surface Coatings 

FTIR measurements were carried out in Attenuated Total Reflectance (ATR) mode using the Nicolet iS10 apparatus with an adapter equipped with a diamond crystal (Thermo Scientific). Measurements were carried out in the mid-infrared range (500–4000 cm^−1^) using: (i) solid samples of nanoparticle powders and (ii) surface modifiers in their pure form. Thermogravimetric analysis (TGA) was carried out using a TGA/DSC3+ thermogravimeter (Mettler Toledo) at a temperature increment 10 °C/min in a temperature window of 25–750 °C and air flow rate 60 mL/min. The total content of C, H, and N elements in modified MNPs was measured using a Vario EL III analyzer with a thermal conductivity detector (Elementar, Langenselbold, Germany) by the accurate weighting of the solid sample and its burning in oxygen at 1150 °C in a dynamic system. Fe, Au, and Si content in solid MNPs was established using the ICP-MS/MS technique, analogically to the methodology presented above (see Section 3.3.2; monitored isotopes: ^28^Si, ^56^Fe, ^197^Au at on-mass mode), but solid samples before digestion were accurately weighted. The oxygen content in the cores of surface-modified MNPs was estimated using the molar ratio of Fe/O in the synthesized bare MNPs as the rest types of the nanomaterials were formed based on such cores (with specific FeO/Fe_2_O_3_ ratio).

### 3.4. Studies on DNA Interactions with Different MNPs Surfaces

To evaluate the efficiency of the DNA adsorption by various magnetic nanoparticles, 200 μL of DNA solutions (0.0786 mg/mL) in media of different salt content: 0.2, 0.5, and 1 M NaCl or MgCl_2_ were prepared. Then, 100 μL of the above solution was mixed with 100 μL aqueous dispersion of magnetic nanoparticles. After 5 min of incubation, the nanosorbent was magnetically collected until the complete separation of nanoparticles was observed. The clear, colorless solution over a precipitate of nanoparticles was taken. In most cases, a separation process was accomplished within less than 1 min, except for nanoparticles Fe_3_O_4_@Au, for which at least 2 min were needed. Then, 150 µL of supernatant was taken and placed in a quartz microcuvette in a Lambda 25 spectrophotometer (Perkin Elmer, Waltham, MA, USA). Then, an absorption spectrum in the 200–350 nm range was captured.

The process of nucleic acids desorption with magnetic nanoparticles was also carried out. Unless otherwise stated, previously separated nanoparticles with adsorbed DNA were suspended in 300 µL of TRIS-EDTA (10 mM TRIS and 0.1 mM EDTA, pH 8). Then the mixture was sonicated for 30 s, left on the magnet to separate nanoparticles, and the supernatant was collected for UV absorption spectrum measurement. The influence of the desorption step duration was examined. For this purpose, 100 µL of the TRIS-EDTA buffer was added to the nanoparticles and left for 30 min. Then, the nanoparticles were separated using a magnet and discarded. Moreover, the influence of temperature on the desorption process was examined. For this purpose, 100 µL of the TRIS-EDTA buffer with nanoparticles was placed in an incubator (Memmert, Büchenbach, Germany) for 30 min at 70 °C. To determine the DNA amount in samples after the adsorption/desorption step, the absorption coefficient was calculated, 2.26 (µg/µL)^−1^·cm^−1^, based on absorbance intensities of the standard curve. The total volumes of adsorption and desorption media (used for calculating total DNA content) were 200 µL and 800 µL, respectively.

## 4. Conclusions

The presented studies carried out a comprehensive, comparative analysis of the impact of Fe_3_O_4_ nanoparticles’ surface modification on their physicochemical properties and DNA binding ability and recovery. A new approach to the co-precipitation method using PEI as a capping agent allowed us to obtain good-quality cationic nanoparticles covered with PEI in a one-step synthesis. The core-shell type nanoparticles synthesized in this way show good stability and shell-dependent magnetism (within limits 44.6–76.8 emu/g Fe, the best for Fe_3_O_4_@APTMS), which allowed them to be used as DNA nano sorbents. The efficiency of surface coating for all MNP types was qualitatively and quantitatively characterized by a combination of TGA, FTIR, ICP-MS/MS, and elemental analysis. Thanks to different interaction mechanisms with DNA, their capacity for reversible DNA binding was characterized. Comprehensive characterization of their morphology, composition, and surface properties has proven effective modification with PEI ligands, Au solid coating, silica, amine-silica, and GO/carboxylated GO decorations. 

Among the examined nanoparticles, the most efficient in adsorption are Fe_3_O_4_@PEI (near 99%), characterized by the electrostatic mechanism of DNA attraction. The most efficient in desorption are Fe_3_O_4_@GOCOOH (near 78%), characterized by the hydrophobic and hydrogen bond-derived mechanism of DNA attraction. The significant influence of the ionic strength and valence of the extraction medium cation (Na^+^ and Mg^2+^) on DNA binding and recovery for the selected nanoparticle types was demonstrated. Based on studies of the desorption process under various conditions, it was concluded that individual selection of desorption conditions for each nanoparticle type is required to obtain high DNA recovery. During the research, it was also found that high EDTA concentration in the medium adversely affects the desorption process and acts as a potential source of interferences during spectrophotometric DNA determination due to iron(III) stripping from MNPs (inducing nanoparticles degradation). Therefore, a medium containing 0.1 mM EDTA was chosen for desorption process investigation in the presented research. These studies drive a new preparation protocol, which should be implemented during the effective design of new magnetic nano sorbents to rapidly isolate nucleic acids from complex samples. 

## Figures and Tables

**Figure 1 ijms-23-08860-f001:**
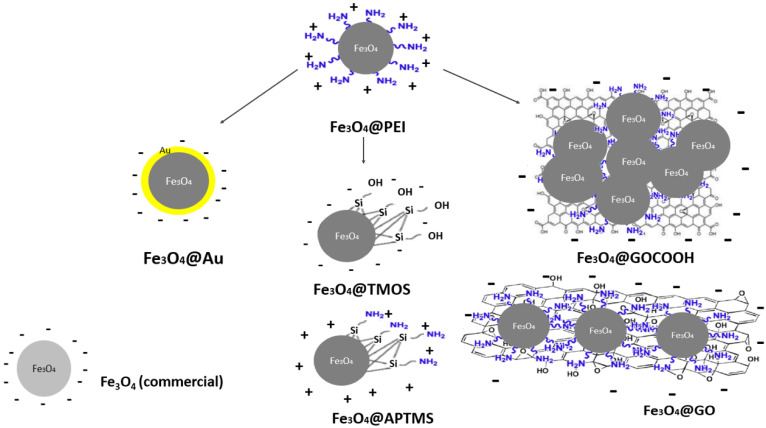
The scheme of magnetic nanoparticles (MNPs) prepared and investigated within this study.

**Figure 2 ijms-23-08860-f002:**
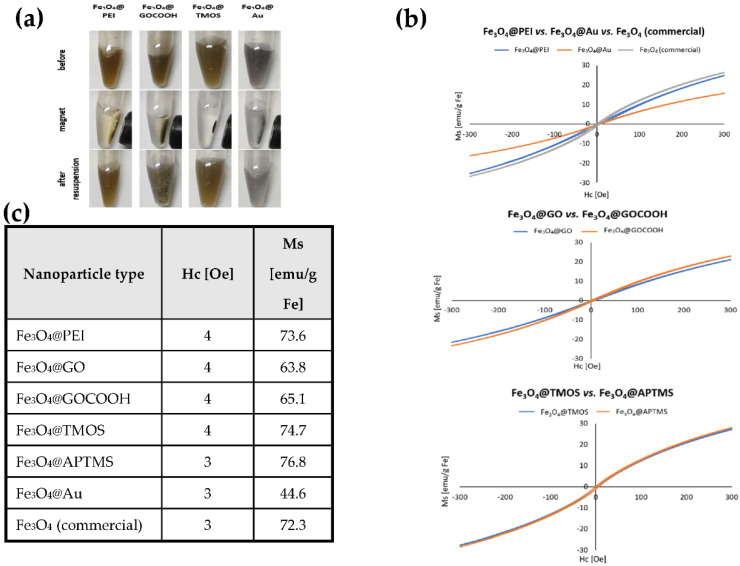
(**a**) Images of the response of chosen MPNs to the external magnetic field and their appearance after brief resuspension, (**b**) magnetization curves of MNPs types obtained by VSM. The external field sweeps from −300 to + 300 Oe. Magnetization units are represented by emu/g Fe, (**c**) calculated values of specific saturation magnetization of MNPs series. Hc—coercivity field, Ms—saturation magnetization.

**Figure 3 ijms-23-08860-f003:**
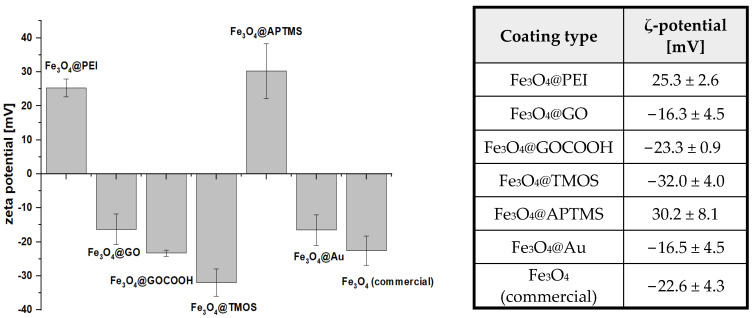
ζ-potential of modified magnetic nanoparticles (*n* = 3).

**Figure 4 ijms-23-08860-f004:**
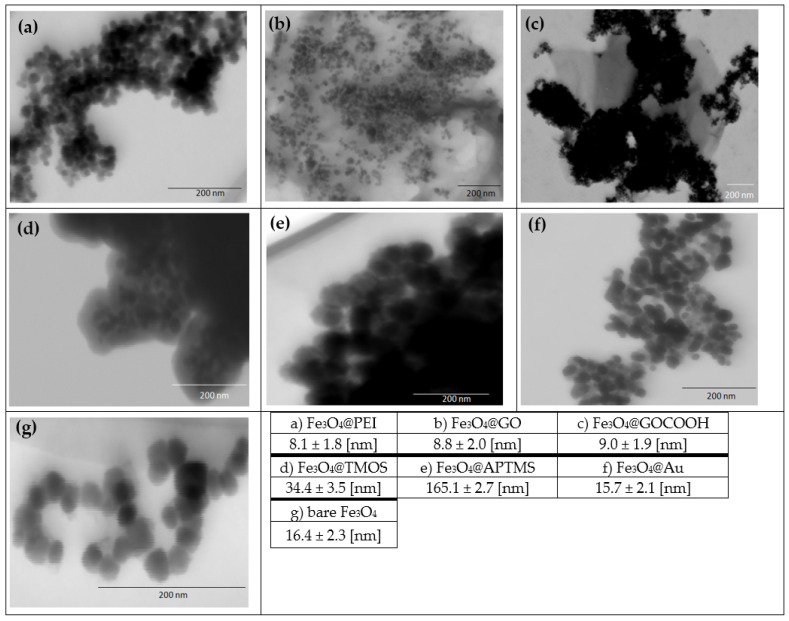
Fragments of STEM images and average sizes of modified, Fe_3_O_4_@PEI-based magnetic nanoparticles (**a**–**f**) and commercial MNPs without polymer shell (**g**) (the average of the measurement results for at least 50 nanoparticles).

**Figure 5 ijms-23-08860-f005:**
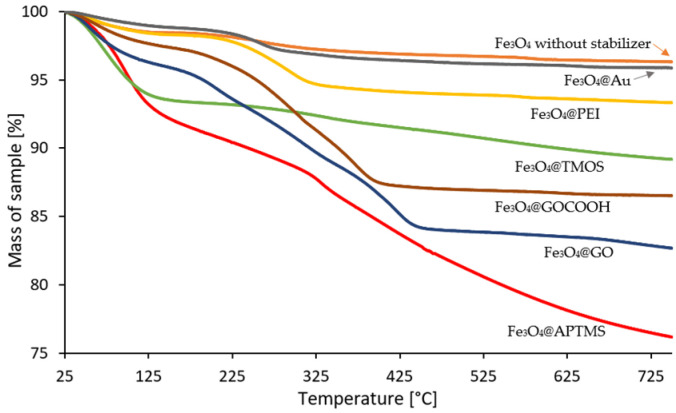
Thermograms of surface-modified magnetic nanoparticles. For comparison, a thermogram of homemade, non-stabilized iron oxide nanoparticles is presented, where no stabilizer was added during co-precipitation (solid orange line).

**Figure 6 ijms-23-08860-f006:**
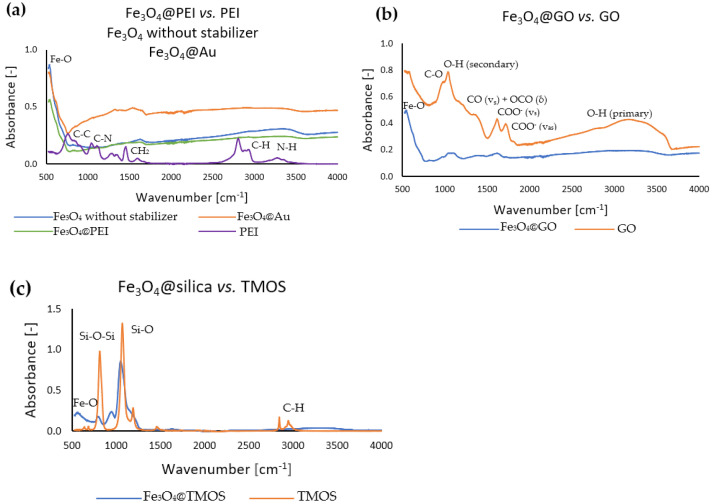
FT-IR absorption spectra of chosen nanoparticle types and corresponding surface modifiers in a pure form: (**a**) Fe_3_O_4_ without stabilizer, Fe_3_O_4_@PEI and Fe_3_O_4_@Au, (**b**) Fe_3_O_4_@GO and pure graphene oxide, (**c**) Fe_3_O_4_@silica and TMOS.

**Figure 7 ijms-23-08860-f007:**
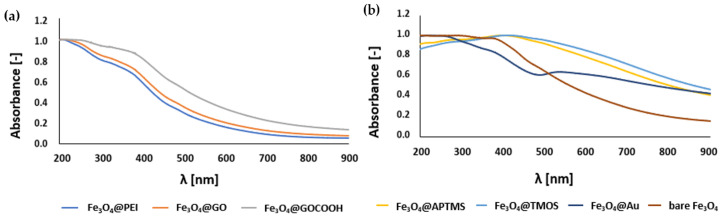
UV-Vis spectra of nanoparticles (**a**) initial Fe_3_O_4_@PEI cores and graphene oxide derivatives-decorated MNPs, (**b**) nanoparticles with solid shell (gold and silica/amine-silica). Spectra were normalized to the maximum value.

**Figure 8 ijms-23-08860-f008:**
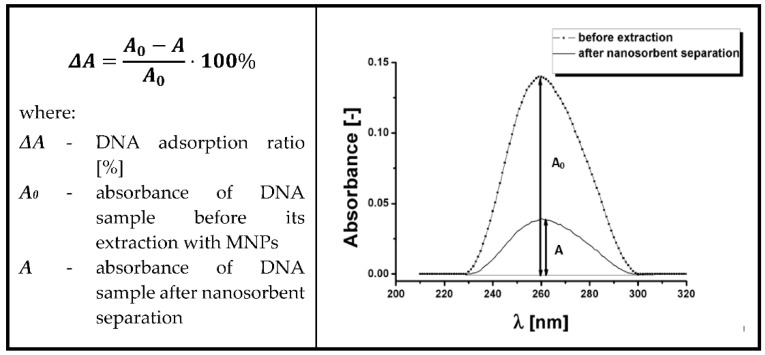
Procedure for evaluation of DNA adsorption efficiency on magnetic nano sorbents. Absorbance peak at 260 nm represents the DNA content in a sample before (dotted line) and after magnetic separation (for better interpretation, obtained spectra were subjected to peak finding in Origin software). The relative adsorption ratio (Δ*A*) is calculated using the given equation.

**Figure 9 ijms-23-08860-f009:**
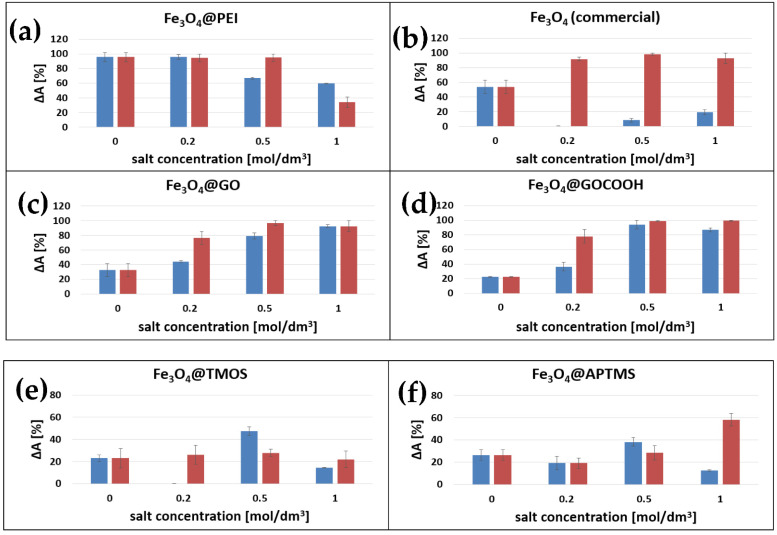
Effect of salt concentration and cation valency (Na^+^—blue bars, Mg^2+^—red bars) on the adsorption ratio of calf thymus DNA (ΔA) on various types of Fe_3_O_4_-based magnetic nano sorbent. Extraction medium composition is an essential factor in DNA binding and subsequent elution from the surface of magnetic nanoparticles. Inefficient adsorption of nucleic acids on nanoparticles may result from electrostatic repulsion between negatively charged nucleic acids and several types of examined nanoparticles. This prompted us to increase the ionic strength of the adsorption medium. The main driving force behind DNA adsorption is the high concentration of ions that reduce the Debye length in the binding solution. This cause effectively shields negative charges and intensely weakens the repulsive electrostatic forces between DNA and nanoparticles.

**Figure 10 ijms-23-08860-f010:**
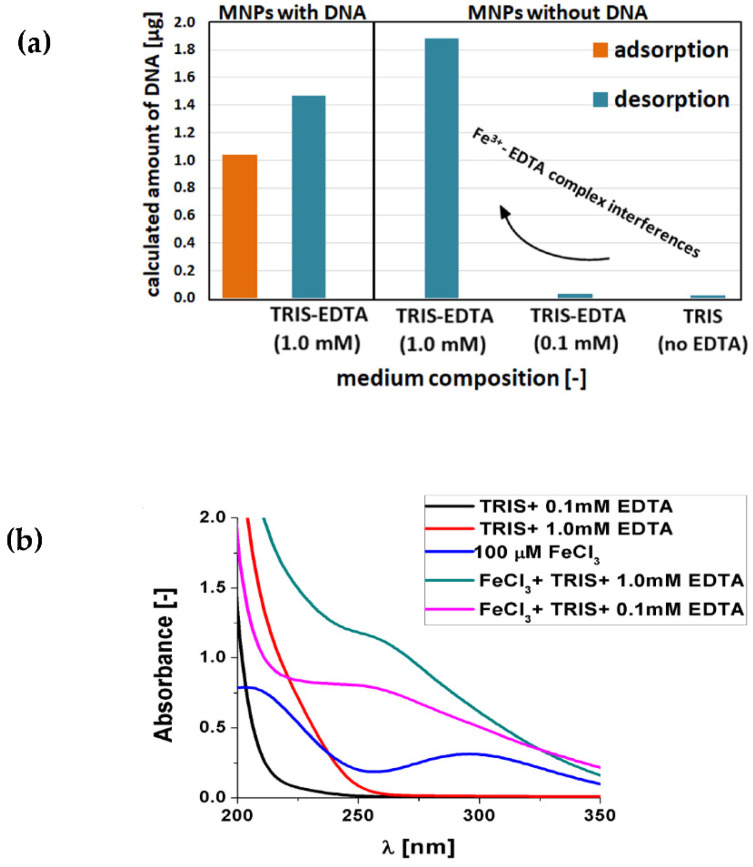
(**a**) The calculated theoretical amount of DNA in desorption media (blue bars) with different EDTA content after incubation with Fe_3_O_4_@PEI MNPs. Orange bars—signal obtained at adsorption step, blue bars—at desorption step. (**b**) Influence of the presence of iron(III) ion on the UV-Vis absorption spectra of individual desorption media with different EDTA concentrations.

**Figure 11 ijms-23-08860-f011:**
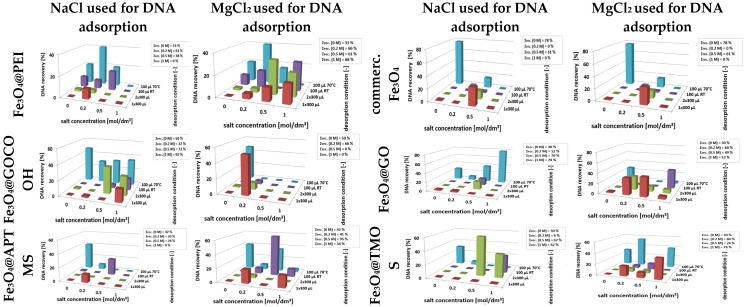
DNA recovery ratios obtained at individual desorption steps. Left column—adsorption recoveries carried out in a solution containing Na^+^ cation, right column—Mg^2+^ cation. Insets show the cumulative recoveries obtained for each magnetic nano sorbent under defined DNA adsorption conditions (salt concentration).

**Table 1 ijms-23-08860-t001:** Calculated compositions of studied MNPs according to ICP-MS/MS and elemental analysis.

Nanoparticle Type	% Fe in Core	% O in Core	% Dominant Hetero-Elements
Fe_3_O_4_@PEI	67.87 ± 0.38	26.85	C-2.29 ± 0.01; N-1.70 ± 0.04
Fe_3_O_4_@GO	61.96 ± 0.22	24.62	C-6.62 ± 0.08; N-1.61 ± 0.08
Fe_3_O_4_@GOCOOH	60.83 ± 0.21	24.17	C-6.39 ± 0.04; N-1.62 ± 0.04
Fe_3_O_4_@TMOS	34.72 ± 0.91	13.79	Si-40.94 ± 0.46; C-0.56 ± 0.06; N-1.09 ± 0.02
Fe_3_O_4_@APTMS	26.19 ± 0.68	10.40	Si-40.94 ± 0.46; C-9.14 ± 0.09; N-3.68 ± 0.04
Fe_3_O_4_@Au	43.21 ± 0.43	17.17	Au-35.83 ± 0.21; C-1.31 ± 0.02
Fe_3_O_4_ without stabilizer	71.57 ± 0.44	28.43	-

**Table 2 ijms-23-08860-t002:** Summary of the amounts of reagents used in the synthesis of nanoparticles.

	Reagent	Weight/Volume Used	Amount [molar]
Fe_3_O_4_ core synthesis	FeCl_3_∙6H_2_O	4.886 g	18.07 mmol
	FeCl_2_∙4H_2_O	2.982 g	15 mmol
	NH_3_∙H_2_O (core synthesis)	15 mL (25%)	218 mmol
	PEI	20 mL (20 mg/mL)	16 μmol
GO/GOCOOH coating	Fe_3_O_4_@PEI MNPs	5 mL (OD_380_ = 1.5) = 4.612 μg/mL	(-) Molar mass unknown
	GO	5 mL (10 mg/mL)	(-) Molar mass unknown
	GOCOOH	5 mL (10 mg/mL)	(-) Molar mass unknown
Silica/amine- silica coating	Fe_3_O_4_@PEI MNPs	150 mL (OD_380_ = 1.5) = 4.612 μg/mL	(-) Molar mass unknown
	NH_3_∙H_2_O (silica coating synthesis)	6 mL (25%)	87.2 mmol
	TMOS (silica coating)	600 μL	4.07 mmol
	TMOS (amine- silica coating)	450 μL	3.06 mmol
	APTMS	150 μL	0.64 mmol
Au coating	Fe_3_O_4_@PEI MNPs	80 mL (OD_380_ = 1.5) = 4.612 μg/mL	(-) Molar mass unknown
	HAuCl_4_∙3H_2_O (Au-seeds)	10 mL (1 %)	0.25 mmol
	Na_3_C_6_H_5_O_7_∙2H_2_O	20 mL (38.8 M)	77.60 μmol
	NaBH_4_	4.5 mL (0.075%)	89.21 μmol
	NaOH	250 mL (10 mM)	25.00 mmol
	HAuCl_4_∙3H_2_O (Au-coating)	100 μL (10 mM)	1 μmol
	NH_2_OH∙HCl	100 μL (30 mM)	3 μmol

## Data Availability

Data available on request.

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
