# Peer review of "Comparative Evaluation of Different Surface Coatings of Fe3O4-Based Magnetic Nano Sorbent for Applications in the Nucleic Acids Extraction"

_ijms, 2022, doi:10.3390/ijms23168860_

Round 1
Reviewer 1 Report
Article (ijms-1835746) provides some fundamental results to improve the extraction of nucleic acid from solution. DNA isolation using magnetic particles is an important area of ​​research in biotechnology and molecular biology. However, in its modern form, the significance of the work is relatively average, since the authors only applied the already known approaches to the isolation of DNA molecules from solution using magnetic particles modified with polyethyleneimine, gold, silicon dioxide, and graphene. Thus, I cannot recommend this study for publication. , it could be accepted for publication after consideration of the following comments:
- Authors should indicate the DNA size and compare their DNA extraction data with already published data for similar systems.
- For comparative studies, commercially available magnetic beads (NucliSENS-easyMAG, BioMérieux, Durham, NC, USA) should be used in accordance with the manufacturer's instructions.
- Evaluation of isolated DNA molecules from various MNPs should be performed using agarose gel electrophoresis.
Author Response
Reviewer 1
Article (ijms-1835746) provides some fundamental results to improve the extraction of nucleic acid from solution. DNA isolation using magnetic particles is an important area of ​​research in biotechnology and molecular biology. However, in its modern form, the significance of the work is relatively average, since the authors only applied the already known approaches to the isolation of DNA molecules from solution using magnetic particles modified with polyethyleneimine, gold, silicon dioxide, and graphene. Thus, I cannot recommend this study for publication. , it could be accepted for publication after consideration of the following comments:
Dear Reviewer, thank You for Your constructive suggestions. However, the main aim of the presented manuscript is not focused on development of new nanomaterials which could be used in nucleic acids extraction (only new nanomaterial investigated in this context was carboxylated graphene oxide which turned out to be slightly more efficient for DNA adsorption than unmodified chemically synthesized graphene oxide). As there are a lot of papers dedicated to above, we decided to focus on the comparison of most popular surfaces used as adsorbents in such a procedure. As the nucleic acids extraction depends on many different factors, one of which, and maybe the most important, is the nanoparticles amount used during DNA extraction. As the concept of this manuscript is based on the comparison of different surface properties on DNA extraction from two different medium compositions, the influence of the nanoparticles amount should be reduced to minimum. To date there is no similar studies presented in the literature. To unify the amount of the nanoparticle cores we used mass spectrometry to evaluate the iron content in the samples after modified magnetic nanoparticles synthesis. Then the obtained modified MNPs were diluted to obtain the unified iron concentration samples, and the same the unified magnetic cores concentration of modified MNPs. This assumption was made as all the modifications were performed with the use of magnetic cores obtained during one synthesis in one batch. That is why it was possible to obtain a series of magnetic nanoparticles with a common feature of magnetic core and its unified amount in the analysed sample, but with very different surface properties. These similarities in the core structure, and differences in surface properties can be, in our opinion, very interesting in the point of view of their influence on the efficiency of nucleic acids extraction in chosen medium composition. We extensively studied the physical properties of obtained modified nanoparticles to exclude any unknown phenomena influence on the MNPs/DNA interaction and evaluation of DNA extraction efficiency. This can be easily observed in desorption experiments, where initially we used very popular and extensively used in the literature desorption medium with few molar, even to 10M, EDTA. In those papers the DNA adsorption and desorption experiments are quantitatively analysed with the use of UV-VIS method with wavelength of about 260 nm. In our experiments we proved, that it may significantly influence the results, because such a big EDTA concentration leads to iron leakage from magnetic cores and significantly increase the light absorption. That is why, in our opinion, this and similar papers, are very important to compare and show as much as possible parameters which can influence the already know, but still not thoroughly studied factors which influence the mechanism and efficiency of nucleic acids extraction with the use of modified nanoparticles.
- Authors should indicate the DNA size and compare their DNA extraction data with already published data for similar systems.
In extraction experiments, we used DNA isolated from calf thymus of ~10 kilo base pair (kbp) [Jumbri K., Kassim M.A., Yunus N.M., et al., Fluorescence and Molecular Simulation Studies on the Interaction between Imidazolium-Based Ionic Liquids and Calf Thymus DNA; Processes (2020), 8(1), 13, DOI: 10.3390/pr8010013]. This, theoretically corresponds to 3.4 mm but according to [Bravo-Anaya L.M., Rinaudo M., Soltero Martínez F.A., Conformation and Rheological Properties of Calf-Thymus DNA in Solution, Polymers 2016, 8, 51; DOI:10.3390/polym8020051] the intrinsic persistence length is found equal to 50 nm. Such an DNA was used in several studies dedicated to development of new magnetic nanomaterials dedicated to nucleic acids extraction. An example of calf thymus DNA as a model compound for DNA extraction studies (both in native and enzyme-digested form) are [Fecková M., Toth J., Šálek P., and et al., Capture of DNAs by magnetic hypercrosslinked poly(styrene-co-divinylbenzene) microspheres; Journal of Materials Science volume 56, pages 5817–5829 (2021), DOI:10.1007/s10853-020-05649-5; Tiwari A.P., Rohival S.S., Suryawanhsi M.V., and et al., Detection of the genomic DNA of pathogenic α-proteobacterium Ochrobactrum anthropi via magnetic DNA enrichment using pH responsive BSA@Fe3O4 nanoparticles prior to in-situ PCR and electrophoretic separation, Microchimica Acta, 83, pages 675–681 (2016) DOI: 10.1007/s00604-015-1710-6; Qi L., Xu R., Gong J., Monitoring DNA adducts in human blood samples using magnetic Fe3O4@ graphene oxide as a nano-adsorbent and mass spectrometry, Talanta 209, (2020), 120523; DOI: 10.1016/j.talanta.2019.120523]. However strict comparison of results between the articles where the main idea was to obtain the best nanoparticles for most efficient DNA extraction procedure with results presented in our manuscript is hard to prepare. This is because we focused on comparison between the influence of chosen surfaces properties (for the first time according to presented idea) on the DNA extraction efficiency. We did not focused on the development of new surface or extraction procedure to obtain the highest efficiency of the DNA extraction.
- For comparative studies, commercially available magnetic beads (NucliSENS-easyMAG, BioMérieux, Durham, NC, USA) should be used in accordance with the manufacturer's instructions.
Nanoparticles with the same mechanism of the DNA interaction as indicated by Reviewer 1 were in the pool of tested nanoparticles and they were silica-coated nanoparticles (Fe3O4@TMOS and amine-silica Fe3O4@APTMS). Due to the huge differences in cores between commercial and produced by us nanoparticles, it is very difficult to compare or plan experiments according to the idea presented by us in our manuscript. However, we believe that nanoparticles coated with silica may simulate commercially available nanobeads as this modification is commonly used in commercial applications.
- Evaluation of isolated DNA molecules from various MNPs should be performed using agarose gel electrophoresis.
Gel electrophoresis might have an advantage over UV-VIS in obtaining a DNA size profile. However, in the context of quantitative results being the part of the idea of the presented research we have abandoned this type of analysis. This is because we need to obtain any quantitative information on the DNA concentration in the sample to be able to quantitively show the influence of surface properties on the DNA extraction. Moreover, we think that gel electrophoresis presents higher detection limits than UV-VIS technique and that is why may have a limited application in presented manuscript.
Reviewer 2 Report
1. The title is not very suggestive.
2. Abstract should clearly inform the important findings in the present study
3. The lengthy sentences may be split in to smaller sentence without change of its meaning.
4. Also, suggested to include the recent references in the introduction part.
5. Several magnetic materials based on iron, cobalt, and nickel have been developed, e.g., magnetic cobalt–zinc ferrite core/SiO2 shell nanosorbents hydrophobic magnetic deep eutectic solvents containing Fe/MnCo/Gd ions, and magnetic ionic liquids including cobalt(II) and nickel(II) complexes.Compare the properties of ferrite obtained from the literature. You can use the references: Journal of thermal analysis and calorimetry 97 (1), 2009, 245-250; Acta Chim. Slov. 2009, 56, 379–385 and Journal of thermal analysis and calorimetry 94 (2), 2008, 389-393
6. The author have detailed the experimental part, in its current form it is not very clear. Insert a table with the number of moles of all reagents used in the synthesis.
7. The authors should correlate their performance results with the already published studies of different researchers to show the priority of their research study.
8 The results and discussions part should be compared with the literature data. To redo the part of results and discussions by a systematic presentation of the results by which the readers of the articles manage to follow the article more easily.
9. Figure 1 is not clear and the writing is very small. To improve.
10. Go through all the hysteresis curves for all the compounds studied in figure 2. They could be superimposed to better see the differences.
11. The TEM images is blurry, to be replaced with one with better resolution. Figure quality is poor throughout. To improve the quality of the figures. Enlarge the characters in the figure 4.
12. To pass a table with mass losses in TGA. Interpretation can be improved by using literature.
13. Add DTA or DSC to complete the thermal analysis part.
14. Not all bands are played in FT-IR. Interpret the other bands, which are treated in the literature.
15. In discussions about structural analysis, XRD and XPS measurements are missing. Let these measurements be added to have a substrate for the correlation of the discussions.
16. Conclusions should be short with important observations.
17. References are not written in unison. Some journals are abbreviated and some are not.
18. The authors must revise language of the manuscript before publication
Round 2
Reviewer 2 Report
I believe that the authors have significantly improved the work and it can be accepted for publication.